# Speech Compressive Sampling Using Approximate Message Passing and a Markov Chain Prior

**DOI:** 10.3390/s20164609

**Published:** 2020-08-17

**Authors:** Xiaoli Jia, Peilin Liu, Sumxin Jiang

**Affiliations:** 1School of Electronic Information and Electrical Engineering, Shanghai Jiao Tong University, Shanghai 200240, China; jxl8601@sjtu.edu.cn; 2School of Electronics and Information Engineering, Shanghai University of Electric Power, Shanghai 200093, China; samjoe_2018@shiep.edu.cn

**Keywords:** compressive sampling, Markov chain, approximate message passing, speech spectrogram, MDCT

## Abstract

By means of compressive sampling (CS), a sparse signal can be efficiently recovered from its far fewer samples than that required by the Nyquist–Shannon sampling theorem. However, recovering a speech signal from its CS samples is a challenging problem, as it is not sparse enough on any existing canonical basis. To solve this problem, we propose a method which combines the approximate message passing (AMP) and Markov chain that exploits the dependence between the modified discrete cosine transform (MDCT) coefficients of a speech signal. To reconstruct the speech signal from CS samples, a turbo framework, which alternately iterates AMP and belief propagation along the Markov chain, is utilized. In addtion, a constrain is set to the turbo iteration to prevent the new method from divergence. Extensive experiments show that, compared to other traditional CS methods, the new method achieves a higher signal-to-noise ratio, and a higher perceptual evaluation of speech quality (PESQ) score. At the same time, it maintaines a better similarity of the energy distribution to the original speech spectrogram. The new method also achieves a comparable speech enhancement effect to the state-of-the-art method.

## 1. Introduction

Compressive sampling (CS) aims to recover a sparse signal from its far fewer samples than that required by the Nyquist–Shannon sampling theorem [1]. The CS procedure can be mathematically expressed as:(1)y=Φx+ω
where *x*
$\in R^{N}$ is the sparse signal, $y\in R^{M}$ is the measurement vector with M≪N, Φ is the sensing matrix. A popular choice of the sensing matrix is the random matrix such as Gaussian matrix. ω is the additive noise. Many CS methods, such as reweighted minimization [2], orthogonal matching pursuit [3], iteratively shrinkage-thresholding algorithm (ISTA) [4] and Bayesian CS [5], can successfully recover sparse signal from the CS samples under certain conditions. In fact, many natural signals are sparse or compressible when represented in a proper basis. This makes CS methods having wide applications in signal sampling and processing. As to speech signals, they are nearly compressible in certain conventional basis, such as fast Fourier transform, modified discrete cosine transform (MDCT) and time–frequency domain [6], and they should be exactly reconstucted from its CS samples theoretically. However, traditional CS methods all can not achieve satisfactory performance when applied to speech signals.

The reason lies in the fact that speech can be seen as a compound signal and is only nearly compressible. In fact, phoneticists usually divide the speech into two categories: voiced sound and unvoiced sound. The voiced sound (e.g., vowel) is produced by the vibration of vocal cord whose spectrum contains only the fundamental tone and its sparse harmonics. The unvoiced sound (e.g., frication) is produced by air turbulence within a human vocal tract whose spectrum is very similar to a white noise and contains low sparsity. Therefore, it is difficult to recover speech from its CS samples. To solve this problem, the structure of the speech spectrogram can be exploited.

In this paper, we propose a CS recovery method which combines approximate message passing (AMP) [7] and Markov chain to exploit the correlation inner and inter the speech frames. Firstly, we employ a Bernoulli–Gaussian mixture to model the marginal distribution of each MDCT coefficient, which is determined by two hidden variables, support and amplitude. The support is the index of the Gaussian function. We assume that the variation of the supports can be characterized by a first-order Markov chain. Secondly, we build a factor graph to conveniently describe the structure of the coefficient matrix and the CS procedure. By means of message passing on the factor graph [8], we can infer the distribution of each MDCT coefficient from CS samples. The inference process can be described as follows. It first estimates the distribution of the MDCT coefficients within each frame from the measurement vector by means of AMP and then passes the message to its hidden variables. Then, the belief propagates within the support matrix in a specific order [9], updating the value of each hidden variable. The belief propagation within the support matrix has exploited the correlation between neighboring elements in the coefficient matrix. At last, the message from the hidden variable is passed back to the corresponding MDCT coefficient, updating the marginal distribution of each coefficient. Then the AMP is executed in each frame again. The whole process will be iterated several times until a stopping condition is met. This alternately iterative process can be refered to as turbo message passing. By this method, we not only exploit the temporal correlation of voiced sound but also make full use of the correlation of the MDCT coefficients in the unvoiced sound frame, leading to an obvious performance improvement compared to traditional CS methods.

The main contributions of the paper can be highlighted as follows: (1) We exploited the structure of speech spectrogram to improve the recovery from CS samples. Both the temporal correlation for the voiced sound and the correlation between neighboring MDCT coefficients in the unvoiced sound frame have been utilized in a turbo message passing framework. A stopping condition in turbo iteration is proposed to guarantee a reliable recovery of speech signal. (2) This method can be applied to speech enhancement and achieve a satisfactory performance.

## 2. Related Work

In this section, we introduce related CS methods. Some of them can be employed to model the temporal correlation of the voiced sound, others can be employed to model the correlation between neighboring MDCT coefficients in the unvoiced sound frame.

The harmonics of voiced sound may span within a few neighboring frames and show strong temporal correlation. In fact, the problem of recovering vectors with temporal correlation from its CS samples is referred to as the multiple measurement vector (MMV) problem. To solve this problem, lots of methods are developed. Lee et al. proposed a SA-MUSIC algorithm to estimate the indexes of nonzero rows when the row rank of the unknown matrix is defective [10]. However, it is apt to allocate all the signal energy to low-frequency components when applied to a speech signal. Vaswani et al. proposed a modified-CS algorithm to recursively reconstruct time sequences of a sparse spatial signal [11]. They used the support of the previous time instant as the “known” part for the current time instant. This algorithm can exploit the temporal correlation between neighboring frames. However, it runs slowly, because the current frame cannot be processed until its previous frame has been reconstructed. Ziniel et al. proposed the AMP-MMV algorithm that assumes the vectors in the MMV problem share common support and the amplitude corresponding the same support is temporally correlated [12]. This temporal correlation is modeled as a Gaussian–Markov procedure. However, when applied to the speech signal, the assumption of common support is not always true. In fact, this is the dynamic CS problem, where the support of unknown vectors changes slowly. To solve this problem the authors proposed DCS-AMP algorithm [13]. They modeled the changes in the support over time as a Markov chain. This algorithm is adequate for the reconstruction of the voiced sound from CS samples, but not for the unvoiced sound. Apart from these methods, temporal correlation can be exploited by another model, e.g., the linear prediction model. Ji et al. proposed to leverage both the speech samples and its linear prediction coefficients jointly to learn a structured dictionary [14]. Based on this dictionary, a high reconstruction quality and a fast convergence can be achieved. However, a training dataset is needed and the dictionary is ad hoc.

The spectrum of the unvoiced sound usually contains smaller and denser frequency components and at the same time maintains a shorter duration, as a result, the temporal correlation among neighboring frames is weaker. The variation range of its amplitude is smaller as well. To exploit the structure of the unvoiced sound, several statistical models have been proposed. Févotte et al. [15] modeled the transient part (e.g., attacks of notes) of musical signal as a Markov chain along the frequency axis. They used a Gibbs sampler to learn the model parameters and finally reduced the “musical noise”. The similarity of neighboring supports in a Markov chain is called persistency in the paper. We adopted this concept in this paper. Jiang et al. [16] modeled the persistency of supports in each frame of speech signal as a Markov chain. They showed that the exploitation of the dependence between neighboring frequency components has improved the reconstruction quality. However, they ignored the temporal correlation between neighboring frames found in the spectrogram. Som et al. further modeled the relationship between neighboring elements in a two-dimensional unknown matrix [17] as a Markov random field. However, due to its simplistic assumption on the probability density function of each element, this method only achieved a little better reconstruction quality than conventional methods.

## 3. Speech Signal Model

In this section, we first approximate the marginal distribution of each MDCT coefficient of speech signal as a Bernolli–Gaussian-mixture in Section 3.1, and then model the correlation between neighboring support variables in the spectrogram as a Markov chain in Section 3.2.

The speech signal is first transformed into a coefficient matrix X${}_{N\times T}$ by a windowed MDCT transform. Wilson or normal window is chosen to generate WMDCT basis. Then each column vector *x_t_*, which is composed of MDCT coefficients in frame *t*, is multiplied by a partial Fourier matrix $\Phi_{M\times N}$. This is the CS procedure in the MDCT domain, which can be mathematically expressed as:(2)$$y_{t}=\Phi x_{t}+\omega_{t}$$
where *y_t_* is the measurement vector, each element of which is a CS sample. The ratio M/N can be defined as the undersampling ratio.

### 3.1. The Marginal Distribution of the Coefficient

We approximate the marginal distribution of each nonzero MDCT coefficient by a Gaussian-mixture, and represent zero coefficient using Dirac function. Then, the probability density function of each MDCT coefficient can be expressed as:(3)$$P\left(x_{n,t}\right)=\omega_{0}\delta \left(x_{n,t}\right)+\sum_{l=1}^{L}\omega_{l}N\left(x_{n,t},\mu_{l},\sigma_{l}^{2}\right)$$
where subscript *n* ∈ [1, …, *N*] index frequency bins, subscript *t* ∈ [1, …, *T*] index frames; $\omega_{l}$, $\mu_{l}$ and $\sigma_{l}^{2}$ denote the weight, mean and variance of the *l*-th Gaussian function, respectively. We suppose that the number of Gaussian functions is *L*. The weights satisfy the normalizing constraint $\sum_{l}^{L}\omega_{l}=1$. These hyper-parameters can be learned from measurement vectors using the expectation maximization (EM) algorithm [18].

### 3.2. The Relationship between Neighboring Coefficients

We assume that each nonzero coefficient *x${}_{n,t}$* is drawn from one of the *L* Gaussian distributions. This distribution can be indexed by a nonnegative integer s${}_{n,t}$ = *l*, (l≤L). When *x*${}_{n,t}$ is zero, *s*${}_{n,t}$ is set as 0. So *s*${}_{n,t}$ indicates whether a coefficient is nonzero and can play the role of the support variable. Depending on the support, *x${}_{n,t}$* can be expressed as: $$x_{n,t}=\left\{\begin{array}{ll} 0 & s_{n,t}=0\\ \theta_{l} & s_{n,t}=l \end{array}\right.$$
where $\theta_{l}$ is the amplitude of *l*-th Gaussian function. So, the MDCT coefficient can be determined by support variable and amplitude variable.

The persistency of MDCT coefficients means that the neighboring coefficients may share the same support variable. This persistency can be modeled as a first order Markov chain. Specifically, for a frequency index *n*, the vector [*s${}_{n,1}$, s${}_{n,2}$, …, s${}_{n,T}$*] across all frames forms a Markov chain. A Markov chain can be used to describe the slow changes in neighboring supports. This change can be described by transition probability, which can be defined as, e.g., P(s${}_{n,t}=0$|s${}_{n,t-1}=l$). This transition propability and the steady-state distribution P(s${}_{n,t}=l$) can fully describe the Markov chain. These hyper-parameters of Markov chain can be learned using EM algorithm.

## 4. The Factor Graph and Reconstruction Algorithm

In this section we demonstrate the CS procedure and the structure of the support matrix on the factor graph in Section 4.1. Then we detail the inference of the speech MDCT coefficients in the turbo message passing framework in Section 4.2.

### 4.1. The Factor Graph of Signal Model

The primary goal of this paper is to estimate each *x${}_{n,t}$* given the observation matrix Y. If we had access to the probability density function of each *x${}_{n,t}$*, we can get an MMSE (minimum mean squared error) estimate of the coefficient. This probability density function can be approximately expressed as posterior probability P(*x${}_{n,t}$*|Y). Calculation of this posterior probability needs to do integration over all other variables in X. That is infeasible for problems of high dimension. Instead, we infer P(*x${}_{n,t}$*|Y) on the factor graph using message passing. The factor graph of the signal model can be illustrated as Figure 1.

Each element in observation matrix Y is denoted by *y${}_{m,t}$*. In the factor graph, round nodes denote variables (*x${}_{n,t}$*, *s${}_{n,t}$* and $\theta_{n,t}$), square nodes denote functions (*f${}_{n,t}$*, *d${}_{n,t}$*, *g${}_{m,t}$*, and *h${}_{n,t-1,t}$*). The function nodes *f${}_{n,t}$*, *d${}_{n,t}$*, *g${}_{m,t}$* represent the marginal distribution of *x${}_{n,t}$*, $\theta_{n,t}$ and *y${}_{m,t}$*. The function node *h${}_{n,t-1,t}$* represents the transition probability between the support variables *s${}_{n,t}$* and *s${}_{n,t-1}$* across neighboring frames for a given frequency bin *n*. MDCT coefficients in each frame are multiplied by sensing matrix Φ to get the CS samples. This procedure is illustrated on each sub-graph, which is surrounded by square frame, of Figure 1. These sub-graphs are stacked one after another and only the first one is fully displayed. Function node and its variable node are directly connected. Variable nodes with dependence are connected via function node. For example, each *x${}_{n,t}$* can be determined by *s${}_{n,t}$* and $\theta_{n,t}$. The three variables are connected through the function node *f${}_{n,t}$* as shown in Figure 1. In the former section, we model the persistency of the supports for a given frequency bin *n* as a first-order Markov chain. We enclose the support matrix within the shadow part of Figure 1 and show the detail of the Markov chain in Figure 2. In the following section, we detail the reconstruction algorithm that combines the approximate message passing in each sub-graph with belief propagation in the support matrix.

### 4.2. Reconstruction Algorithm

In each sub-graph of Figure 1 we need to estimate original MDCT coefficient vector *x${}_{t}\in$ C^N^* in frame *t* from its measurement vector *y${}_{t}\in$ C^M^*. Due to dense connections between *x${}_{t}$* and *y${}_{t}$*, there exist many short cycles in the factor graph. It is not a good choice to infer the marginal distribution of each MDCT coefficient using sum-product algorithm. Because it requires that the factor graph is cycle-free [8]. Instead, we estimate the marginal distribution using AMP algorithm, which can effectively recover a sparse signal from CS samples when the measure matrix is not sparse [19]. After initializing the probability density function of each MDCT coefficient, we run AMP algorithm and get an approximation of the marginal distribution *f${}_{n,t}$*. The marginal distribution can be referred to as a message. AMP is run simultaneously for each frame. Then messages about *f${}_{n,t}$* is passed on to its support variable and amplitude variable. This procedure can be named as observation sector.

After receiving the message from *f${}_{n,t}$*, variables *s${}_{n,t}$* and $\theta_{n,t}$ update their marginal distributions respectively. To exploit the relationship between neighboring elements in the support matrix, we will do belief propagation along the row and column of the support matrix in sequence. First, belief propagation is executed for each row from the first frame to the last frame as shown in Figure 2 where the direction of the message passing is not shown. Belief propagation along the row of the support matrix has exploited the temporal correlation of the voiced sound. Second, belief propagation is executed for each column from the lowest frequency bin to the highest frequency bin as shown in Figure 3. It can exploit the persistency of support variables for the unvoiced sound. Then, these two steps are repeated but in a backward direction, that is from the last frame to the first frame and from the highest frequency bin to the lowest frequency bin. During the belief propagation, the support about each *s${}_{n,t}$* has been updated. Theoretically, a Markov chain can model the dependence of neighboring supports only along a specific axis. Considering the fact that the voiced sound accounts for the majority of speech energy, and that the voiced sound is indispensable to speech intelligibility, we assumed a Markov chain prior on the row of support matrix. Transition probability of the Markov chain can be learned using the updated support variables. Belief propagation along the column in the procedure can provide additional structure information of unvoiced sound. This may improve the quality of recovered speech. We did not exploit the structure of the amplitude matrix to avoid the over-smoothing of recovered speech [20]. At last the message about each *s${}_{n,t}$* is passed on to its function node *f${}_{n,t}$*. This procedure can be named as pattern sector. Then, we restart the observation sector using the updated message of each *x${}_{n,t}$*. This is the entire procedure of the turbo message passing.

Turbo iteration continues until a stopping condition is met or the number of iterations is larger than a preset integer. The index of turbo iteration can be denoted by *k* in this paper. The stopping condition is defined as $\|X^{\left(k\right)}-X^{\left(k-1\right)}\|<\|X^{\left(k+1\right)}-X^{\left(k\right)}\|$. Here, X${}^{\left(k\right)}$ denotes the estimate of MDCT coefficient matrix in *k*-th turbo iteration. This condition can avoid the divergence of the proposed algorithm. In fact, without this stopping condition, the belief propagation along the Markov chain would allocate signal energy to frequency components that should be zero as the number of turbo iterations increases. This should be avoided since it will cause severe distortion.

After the turbo iteration, an inverse MDCT transform is performed on the coefficient matrix to obtain the recovered speech signal.

The whole flowchart of the proposed algorithm is shown in Figure 4.

To detail the message flow in the factor graph, several microstructures are depicted in Figure 5. In these figures, arrows at the end of edges indicate the direction to which messages are flowing. The notation $\upsilon_{A\to B}$ denotes a message that is propagated from node *A* to *B*. In all figures except the observation sector the message passing occurred according to the sum–product algorithm: the message sent from a function node along an edge equals the integral (or sum) of the product of the local function with messages coming into that node along all other edges; the message comes from a variable node is the product of the messages coming into the variable node along all other edges.

When the message about *x*${}_{n,t}$ is propagated out of the observation sector, it is passed on to the corresponding variables of *s*${}_{n,t}$ and $\theta_{n,t}$. This procedure is illustrated in Figure 5a. These two messages can be computed using the sum–product algorithm as:(4)$$\upsilon_{f_{n,t}\to s_{n,t}}\propto \int_{x_{n,t}}\int_{\theta_{n,t}}f\left(x_{n,t},s_{n,t},\theta_{n,t}\right)\upsilon_{x_{n,t}\to f_{n,t}}\left(x_{n,t}\right)\upsilon_{\theta_{n,t}\to f_{n,t}}\left(\theta_{n,t}\right)$$
(5)$$\upsilon_{f_{n,t}\to \theta_{n,t}}\propto \sum_{s_{n,t}}\int_{x_{n,t}}f\left(x_{n,t},s_{n,t},\theta_{n,t}\right)\upsilon_{x_{n,t}\to f_{n,t}}\left(x_{n,t}\right)\upsilon_{s_{n,t}\to f_{n,t}}\left(s_{n,t}\right)$$
where ∝ denotes equality up to a scaling factor. As to the message passing in the shadow part of Figure 1, there are two cases. When the message passing occurred along the time axis, the message passed on from variable node *s${}_{n,t}$* to function node *h${}_{n,t,t+1}$* can be expressed as:
(6)$$\upsilon_{s_{n,t}\to h_{n,t,t+1}}\propto \upsilon_{h_{n,t-1,t}\to s_{n,t}}\upsilon_{f_{n,t}\to s_{n,t}}$$
and a message passed on from function node *h${}_{n,t,t+1}$* to variable node *s${}_{n,t+1}$* can be expressed as:
(7)$$\upsilon_{h_{n,t,t+1}\to s_{n,t+1}}\propto \sum_{s_{n,t}}h_{n,t,t+1}\upsilon_{s_{n,t}\to h_{n,t,t+1}}$$

These procedures are shown in Figure 5b. The message passing along the frequency axis can be defined in the same way.

According to the sum–product algorithm, the message from *f${}_{n,t}$* to *x${}_{n,t}$* can be calculated as in Equation (Equation 9):
(8)$$\upsilon_{s_{n,t}\to f_{n,t}}\propto h_{n-1,n,t}h_{n,n+1,t}h_{n,t-1,t}h_{n,t,t+1}$$
(9)$$\upsilon_{\theta_{n,t}\to f_{n,t}}\propto d_{n,t}$$
(10)$$\upsilon_{f_{n,t}\to x_{n,t}}\propto \sum_{s_{n,t}}\int_{\theta_{n,t}}f_{n,t}\upsilon_{\theta_{n,t}\to f_{n,t}}\upsilon_{s_{n,t}\to f_{n,t}}$$
where messages $\upsilon_{s_{n,t}\to f_{n,t}}$ and $\upsilon_{\theta_{n,t}\to f_{n,t}}$ have been given too. These procedures can be shown as the Figure 5c,d.

## 5. Result and Discussion

In this section, we reconstruct speech signal from CS samples using the proposed algorithm, and make a comparison with other algorithms, including DCS-AMP [13], EM-GM-AMP [18], MRF [17] and FISTA [21]. DCS-AMP models the marginal distribution of each MDCT coefficient as a Bernoulli–Gaussian-mixture and the persistency of supports along the time axis as a Markov chain. It aims to recover sparse, slowly time-varying vectors from CS samples. EM-GM-AMP models the marginal distribution of each nonzero coefficient as a Gaussian-mixture and learns hyper-parameters using EM algorithm. It does not exploit the signal structure information. The MRF method models the marginal distribution of each coefficient as a Bernoulli–Gaussian and the structure of the support matrix as a Markov random field. FISTA is a modification of ISTA [4]. It makes a compromise between the squared error and signal sparsity. It does not exploit the signal structure information as well.

In all experiments, the speech signal is partitioned into frames using the same window function. The neighboring frames have a 50% overlap. The sensing matrix is a time-invariant, partial Fourier matrix Φ.

First, we evaluate the reconstruction performance by a quasi-SNR (signal-to-noise ratio):(11)$$\mathrm{SNR}\left(x,\hat{x}\right)=10\log_{}\frac{{\left\|x\right\|}_{2}^{2}}{\|x-\hat{x}{\|}_{2}^{2}}$$
where *x* and $\hat{x}$ are original speech and the reconstructed speech, respectively. We reconstruct three speech excerpts, including an adult male’s voice, adult female’s voice and child’s voice, from the CS samples. Each speech excerpt is recorded using the cellphone in a quiet environment. The duration of each speech excerpt is 25 s. Figure 6 shows the SNR under different undersampling ratios for the three speech excerpts. Each result is an average of 50 Monte Carlo experiments.

Obviously, as the undersampling ratio increases, the SNR increases too. In general, the reconstruction performance improves as the signal model becomes more accurate. While DCS-AMP and EM-GM-AMP offer better recovery performance than FISTA and MRF, the proposed algorithm achieves the best performance under nearly all undersampling ratios. However, when the undersampling ratio dropped to 0.2, the proposed algorithm was exceeded by DCS-AMP. This may be explained that when the measurement information about broadband signal is severely lacking, the estimate from CS samples using AMP is not accurate enough, so the belief propagation using the estimate does not improve the reconstruction performance. The SNR gap between the proposed algorithm and DCS-AMP is lager for the adult female’s voice than the adult male’s voice. This may be explained by the fact that the female’s pitch is higher than male, and more signal energy is located in higher frequency components. This energy distribution can be captured by the belief propagation along the frequency axis in the spectrogram. When more measurements are provided, the EM-GM-AMP exceeds the DCS-AMP for the adult female’s voice at undersampling ratio 0.4 or larger. This can be explained by the fact that an adult female’s voice changes quickly, so the temporal correlation is getting smaller. For the child’s voice, the difference in the SNR for the three algorithms (the proposed, DCS-AMP and EM-GM-AMP) is small especially when the undersampling ratio is large. This may be explained by the child being too young and his pronunciation not fluent. The Markov chain that models the temporal correlation in the proposed method and DCS-AMP does not work well. Despite this, the two methods still achieved higher SNR than EM-GM-AMP. MRF and FISTA algorithms achieved lower SNR than the aforementioned methods. MRF method models the marginal distribution of each coefficient as a Bernoulli–Gaussian. This distribution is compatible with the Ising model assumed on the support matrix. However, this marginal distribution function is too simplistic. The SNR is lower than other methods. FISTA method can be adjusted by a parameter λ. The larger the λ, the more sparse the solution. In the experiments, λ is set as 0.01. Since this method does not exploit the signal structure, the SNR gap between the first three methods and the FISTA gets larger as the undersampling ratio increases.

Second, we compare the spectrograms of reconstructed speeches by different algorithms. Figure 7 shows the spectrogram of a speech excerpt, and the counterparts of reconstructed speech excerpts. The undersampling ratio is set as 1/3. It is obvious that the signal is not sparse enough, its energy is mainly located in low-frequency components. FISTA and EM-GM-AMP do not exploit the structure within the coefficient matrix and allocate too much energy to high-frequency components. MRF is better than these two algorithms from this aspect. In fact, the block sparse structure in the support matrix is captured by the Markov random field. The nonzero coefficients of recovered speech appeared in clusters. This method discards the sporadic coefficients and may distort subjective perception. The spectrograms obtained by DCS-AMP and the proposed algorithm are more similar to the original spectrogram than other algorithms. However, some differences exist. That is, the detail of the structure is more clearly demonstrated on the spectrogram of the reconstructed speech by the proposed algorithm. This means that the proposed algorithm can reduce the additive white Gaussian noise that exists in the original speech signal. This effect may be confirmed by the following experiment.

Next, we give a perceptual evaluation of speech quality (PESQ) [22] scores for different algorithms in Table 1. The PESQ score is used to evaluate the difference between the degraded speech and original speech and estimate the subjective mean opinion score. The higher the score, the better the quality. Here, the degraded speech is the recovered speech from CS samples. In the experiment, the original adult male’s speech and adult female’s speech are chosen from TIMIT acoustic–phonetic continuous speech corpus. The sampling frequency is 16 kHz. The original child’s speech is downsampled to 16 kHz too. The duration of each speech is 25 s. The undersampling ratio is set as 1/3. Each result is the mean of 50 Monte Carlo experiments. As the table shows, the proposed algorithm has achieved the highest PESQ score. There is little difference between the scores obtained by the DCS-AMP and EM-GM-AMP. In general, they have achieved suboptimal results. The FISTA method achieved a better score than the MRF method. This may be explained that the latter method discards some coefficients far away from the clusters described by the Markov random field. It is worth mentioning that the FISTA method has achieved a similar score to EM-GM-AMP method and DCS-AMP method when applied to the adult male’s voice. We speculate that the adult male has a lower pitch, and the most signal energy is located in the voiced sound which can be reconstructed well using conventional CS methods.

Finally, from the result in Table 1, we can conclude that the proposed algorithm has an effect of speech enhancement. So far, we have reconstructed clean speech signals from the CS samples. Afterwards, we did a compressive sampling of the noisy speech and reconstructed it using the proposed method. The noisy speech signals were randomly selected from the NOIZEUS database, including five excerpts of a male voice indexed by 01–05 and five excerpts of a female voice indexed by 11–15. The noisy speech signals contain babble noise at different SNR. We computed the PESQ score and STOI (short-time objective intelligibility) score [23] using the recovered speech and clean speech, which were not corrupted by any noise. Each result is the mean of 50 Monte Carlo experiments. The higher the score the better the enhancing effect. We make a comparison with a state-of-the-art speech enhancement algorithm [24], which we refer to as the reference method. The undersampling ratio is set as 0.9 for both methods. The experimental results are shown in Figure 8 and Figure 9. The scores of the reference method are taken from the published paper. It can be seen from Figure 8 that the proposed method achieves a comparable PESQ score to the reference method for the male voice. As the SNR of the noisy speech increases, the PESQ score of the recovered female voice approximates to the reference method gradually. It can be seen from Figure 9 that as the SNR of the noisy speech increases, the proposed method achieves a higher STOI score than the reference method, both for the male voice and the female voice. Experimental results show that our method achieves comparable PESQ scores and STOI scores to the state-of-the-art speech enhancement algorithm.

## 6. Conclusions

In this paper, we proposed a reconstruction algorithm for speech CS. Firstly, we analyzed the speech spectrogram in detail and found that a certain structure exists in it. That is, the voiced sound shows persistency along the time axis and the unvoiced sound shows persistency along the frequency axis. Considering the fact that the voiced sound takes a major part of signal energy, we assumed a Markov chain along the time axis in the support matrix. Then, we proposed a new method for speech CS reconstruction. In this method, a turbo message passing framework is employed which alternately executes AMP within each frame and belief propagation in the support matrix. The belief propagation can exploit the dependence between the neighboring elements in MDCT coefficient matrix. Additionally, in order to avoid the divergence of the method, we proposed a rigorous stopping condition for the turbo iteration. Experimental results show that the proposed algorithm achieved a higher SNR, a more similar energy distribution to the original spectrogram, and a higher PESQ score than other traditional reconstruction algorithms. We also employ the proposed method to enhance noisy speech signals in the NOIZEUS database and make a comparison with a state-of-the-art speech enhancement method. The results show that the proposed algorithm can achieve a comparable speech enhancement effect. 

## Figures and Tables

**Figure 1 sensors-20-04609-f001:**
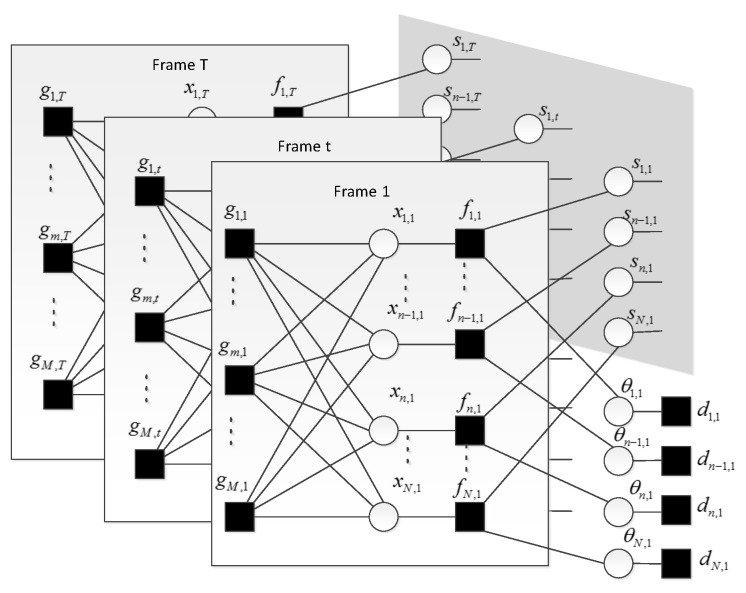
Factor graph presentation of speech signal model. Only the first, the last and a frame indexed by *t* have been shown. Each compressive sampling (CS) sample has been incorporated into the function node. The *f${}_{n,t}$* represents conditional probability P(*x${}_{n,t}$*|*s${}_{n,t}$*, $\theta_{n,t}$). All the support variables have been enclosed in the shadow portion and the connections between them are shown in Figure 2.

**Figure 2 sensors-20-04609-f002:**
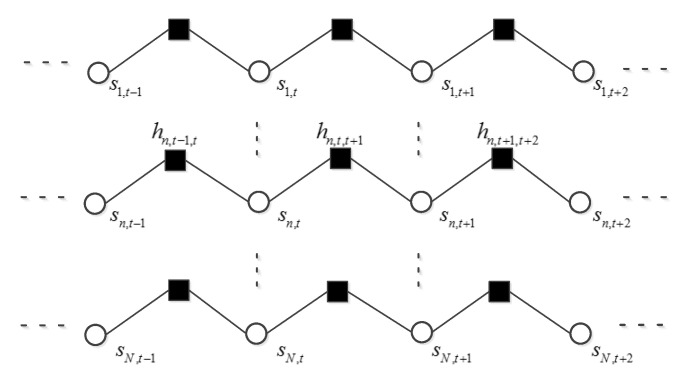
Schematic diagram of the Markov chain, which is composed of the supports corresponding to the same frequency bin *n*. Only the first, the last and the Markov chain *n* are illustrated.

**Figure 3 sensors-20-04609-f003:**
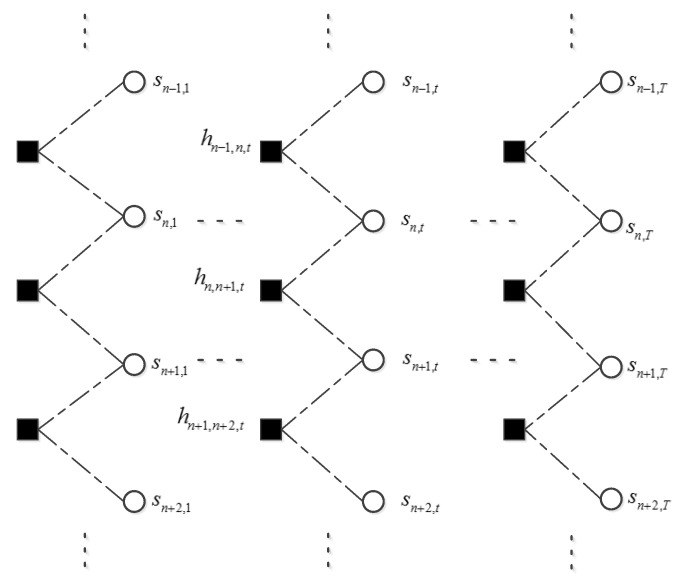
Schematic diagram of belief propagation along frequency axis in each frame. Only the first, the last and the frame *t* are illustrated. The dashed line indicates that this is not a Markov chain. Belief propagation is to exploit the persistency along frequency axis.

**Figure 4 sensors-20-04609-f004:**
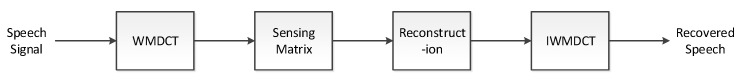
The flowchart of the proposed algorithm.

**Figure 5 sensors-20-04609-f005:**
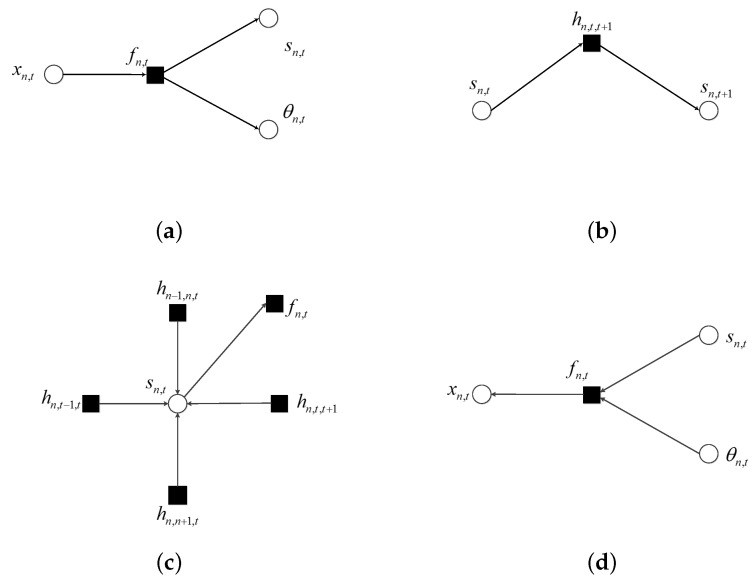
Factor graph for different microstructures: (**a**) message propagated out of observation sector; (**b**) message passing along time axis for frequency bin *n*; (**c**) message from support variable to its function node; (**d**) messages coming into observation sector.

**Figure 6 sensors-20-04609-f006:**
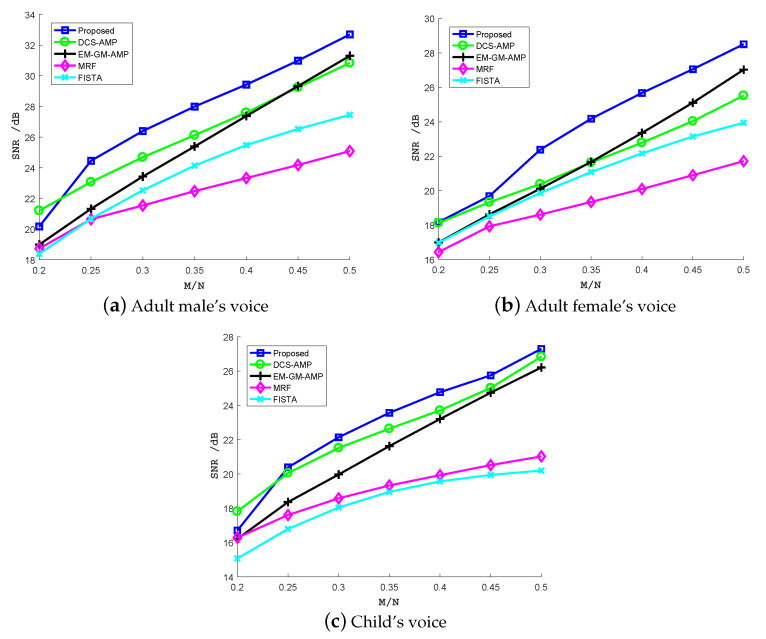
Signal-to-noise ratio (SNR) (dB) of recovered three speech excerpts: (**a**) adult male’s voice, (**b**) adult female’s voice, (**c**) child’s voice, from CS samples using the proposed algorithm, DCS-AMP, EM-GM-AMP, MRF and FISTA under different undersampling ratios.

**Figure 7 sensors-20-04609-f007:**
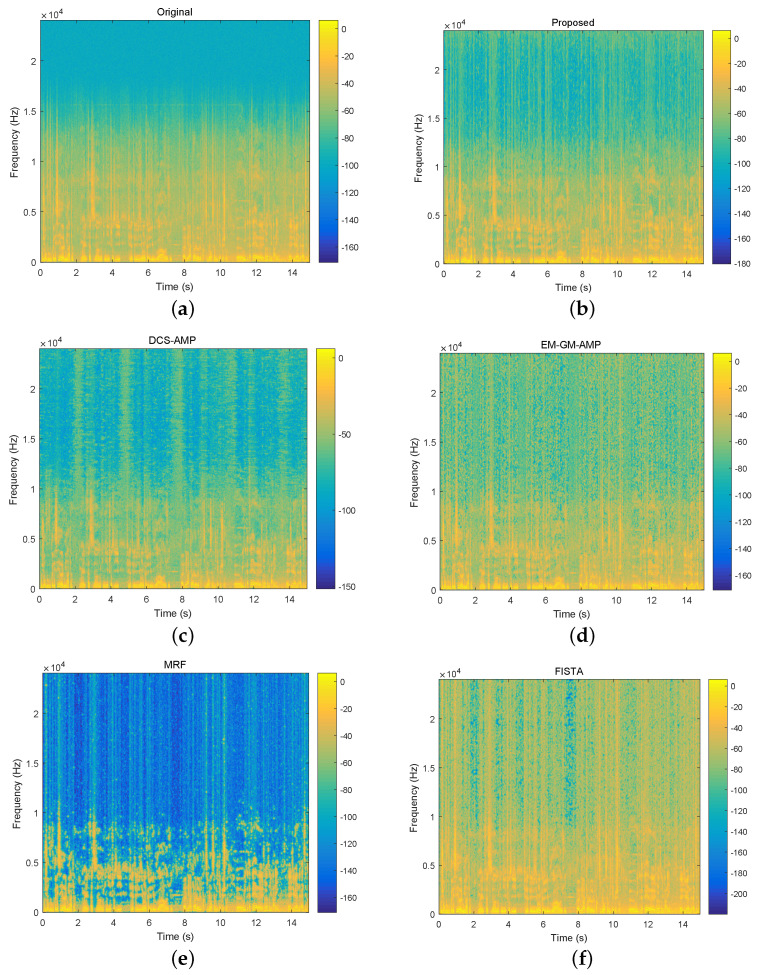
The spectrograms of a 15s speech excerpt (top left) and recovered speech excerpts using different algorithms under undersampling ratio 1/3. The bar on the right of the spectrogram indicates the amount of energy with larger energy in a higher position. (**a**) spectrogram of original speech excerpt; (**b**–**f**) spectrograms of recovered speech excerpts using the proposed algorithm, DCS-AMP, EM-GM-AMP, MRF and FISTA.

**Figure 8 sensors-20-04609-f008:**
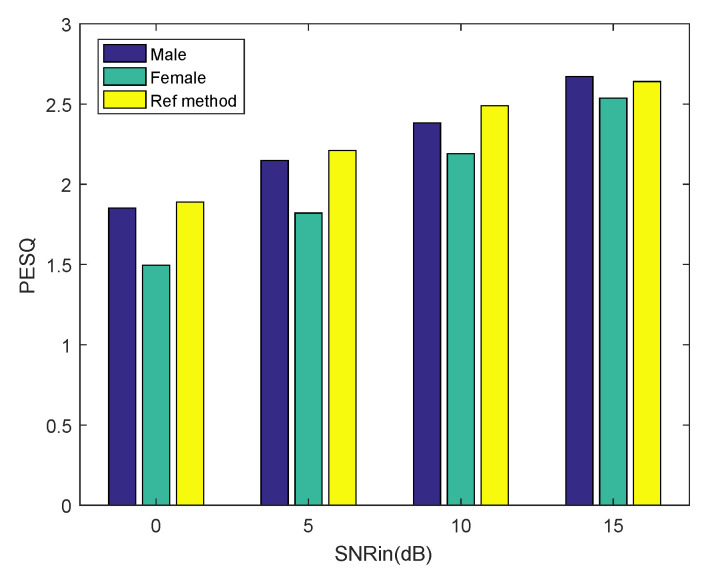
PESQ performance of the proposed method and the reference method in babble noise. We enhance five excerpts of noisy speech under different SNR for the male and female voices.

**Figure 9 sensors-20-04609-f009:**
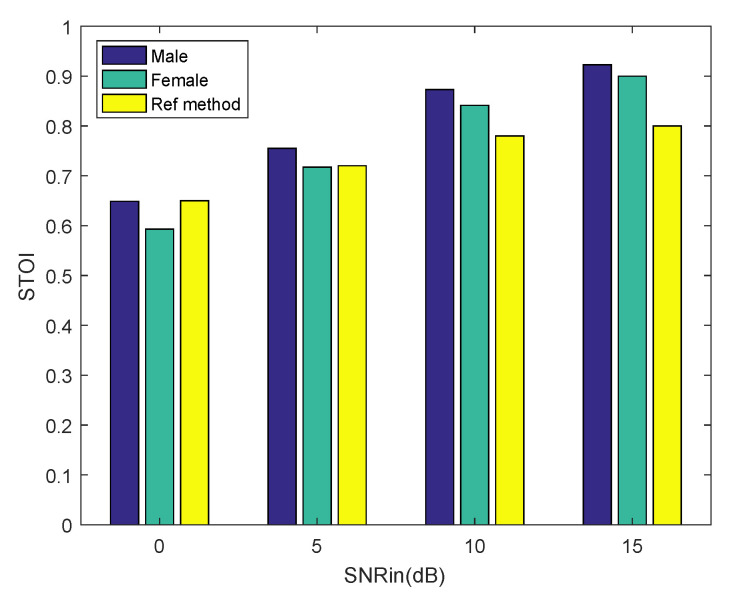
Short-time objective intelligibility (STOI) performance of the proposed method and the reference method in babble noise. We enhance five excerpts of noisy speech under different SNR for male and female voices.

**Table 1 sensors-20-04609-t001:** The perceptual evaluation of speech quality (PESQ) scores of recovered speech from CS samples using different reconstruction algorithms.

Algorithm	Proposed	DCS-AMP	EM-GM-AMP	MRF	FISTA
Male’s voice	3.0974	2.9335	2.8834	2.3534	2.9226
Female’s voice	2.9504	2.7445	2.7273	2.3029	2.6966
Child’s voice	3.706	3.5179	3.588	2.6706	3.1453

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
