# Peer review of "Speech Compressive Sampling Using Approximate Message Passing and a Markov Chain Prior"

_sensors, 2020, doi:10.3390/s20164609_

Round 1

Reviewer 1 Report

The authors address the problem of compressive sampling and signal reconstruction. Authors’ experiments show that, compared to other traditional compressive sampling methods, the proposed method achieved a higher signal-to-noise ratio, maintained a better similarity of energy distribution to the original speech spectrogram, and achieved a comparable speech ehancement effect to the state-of-the-art method.

I believe the reviewed manuscript reports novel results and can be of some significance to the community. Conclusions are supported by the evidence. The manuscript is not very well written, yet the data is appropriately presented. Accordingly, I can recommend the acceptance of the manuscript to be published in Sensors with the following remarks:

  1. There are some outdated references, e.g. dated 1998. Authour should investigate more up-to-date research.
  2. There are numerous spelling and grammar mistakes, e.g. improper use of contractions (such as “isn’t”) instead of full forms, and multiple mistakes in the use of articles. More formal style should be adopted.

Reviewer 2 Report

In this paper, the author proposes a compressive sampling (CS) method for speech signals,  and the methods combines approximate message passing (AMP) and Markov-Chain (MC). Extensive experiments are conducted to demonstrate the effectiveness of proposed method.

the research appears to be sound, the results are exciting for their potential as a research tool, the study will be of interest to Sensors readers, and I believe that the paper merits publication in Sensors.

However, I do have some comments given as follows:

  1. Section 2, why the MDCT is selected but not other transform? It can be helpful to highlight the reason.
  2. Figure 2 should be enlarged for better visualization.
  3. The resolution of all figures are quite low.
  4. The paper should be double-checked for adherence to the format requirements for reference citation, including the use of square brackets vs. rounded brackets.

Reviewer 3 Report

The paper presents a method to reconstruct the speech signal from CS sample. The paper is interesting, however it is not always easy to read. At the beginning of each section there should be an explanation of what will be presented in the section and / or in its subsections. I suggest to reduce the introduction and add a section dealing with the state of the art being careful to maintain the logical thread between the proposed methods, highlighting their advantages and disadvantages. CS sampling method should be explained at the beginning of the state of art to better understand its implementation in the different methods. There are many acronyms, it must be checked that all are correctly defined the first time they appear in the text. The originality, usefulness and importance of the scientific contribution proposed in the paper should be highlighted more.
